# SETDB1-Mediated Silencing of Retroelements

**DOI:** 10.3390/v12060596

**Published:** 2020-05-30

**Authors:** Kei Fukuda, Yoichi Shinkai

**Affiliations:** Cellular Memory Laboratory, RIKEN Cluster for Pioneering Research, RIKEN, Wako 351-0198, Japan

**Keywords:** SETDB1, heterochromatin, H3K9me3

## Abstract

SETDB1 (SET domain bifurcated histone lysine methyltransferase 1) is a protein lysine methyltransferase and methylates histone H3 at lysine 9 (H3K9). Among other H3K9 methyltransferases, SETDB1 and SETDB1-mediated H3K9 trimethylation (H3K9me3) play pivotal roles for silencing of endogenous and exogenous retroelements, thus contributing to genome stability against retroelement transposition. Furthermore, SETDB1 is highly upregulated in various tumor cells. In this article, we describe recent advances about how SETDB1 activity is regulated, how SETDB1 represses various types of retroelements such as L1 and class I, II, and III endogenous retroviruses (ERVs) in concert with other epigenetic factors such as KAP1 and the HUSH complex and how SETDB1-mediated H3K9 methylation can be maintained during replication.

## 1. Introduction

Transposable elements (TEs) comprise more than 40% of most extant mammalian genomes [1,2]. Among these, retroelements including short/long interspersed elements (SINEs/LINEs) and endogenous retroviruses (ERVs) are still active and capable of retrotransposition [3,4]. Although retrotransposition contributes to genome diversification and evolution, it can cause genome instability, insertional mutagenesis, or transcriptional perturbation and is often deleterious to host species [5,6,7]. Therefore, evolution has also driven the development of multiple defense mechanisms against retrotransposition. The first line of defense is transcriptional silencing of integrated retroelements, using various epigenetic modifications, such as DNA methylation and histone H3 lysine 9 tri-methylation (H3K9me3) [8,9]. Since we identified H3K9 methyltransferase SETDB1 as a pivotal factor for retroelement repression in mouse embryonic stem cells (mESCs) [10], significant efforts have been undertaken to elucidate the mechanism of SETDB1-mediated retroelement silencing. This review highlights recent advances in our understanding of SETDB1-mediated retroelement silencing and the regulation of SETDB1 itself.

## 2. Regulation of Retroelement Expression by H3K9 and DNA Methylation

TEs are major factors in genome evolution and have helped shape the structure and function of many genes [5]. However, TEs must be controlled by host cells because they can cause genomic instability. TEs are roughly classified into DNA transposons and retroelements. DNA transposons move by a “cut and paste” mechanism. About 3% of the human genome consists of DNA transposon derived sequences, but active DNA transposons do not exist anymore in the human or mouse DNA. Retroelements are also mobilized by a “copy and paste” mechanism, involving reverse transcription of a RNA template into DNA, followed by integration in the host genome. There are two classes of retroelements: long terminal repeat (LTR) retrotransposons and non-LTR retrotransposons. The former includes ERVs which are relics of ancient retroviruses integrated into germline DNA. ERVs are abundant in mammals, comprising approximately 10% of the mouse genome and 8% of the human genome [1,2]. ERVs are subdivided into three classes (class I, II, III) based on the similarity of their reverse transcriptase genes or their relationship to different genera of exogenous retroviruses [11]. While there are no known human ERVs capable of autonomous-replication and transposition, the mouse genome contains replication-competent copies of class I and II ERVs that are responsible for 10–12% of all published germline mutations by mediating new transposition events [3]. Non-LTR retrotransposons are as old as the earliest multi-cellular organisms. In mammals, they consist mostly of LINEs and SINEs. LINE-1s (L1s) comprise about 17% of the mouse and human genome, and are the only autonomous mobile DNA elements that are currently active in humans. L1s have also been responsible for genomic insertion of over a million non-autonomous SINEs [12]. Retrotransposition can cause genome instability, insertional mutagenesis, or transcriptional perturbation and is often deleterious to host species [5,6,7]. Therefore, evolution has also driven the development of multiple defense mechanisms against retrotransposition, including transcription silencing of the integrated retroelements. Methylation of the fifth carbon of cytosines (5-methylcytosine) is one of the most characterized repressive epigenetic modifications in mammals. DNA methylation has been implicated in the classic epigenetic phenomena of genomic imprinting [13,14,15,16] and X-chromosome inactivation [17,18]. DNA methylation is catalyzed by the de novo DNA methyltransferases DNMT3A and DNMT3B and the maintenance DNA methyltransferase DNMT1. *Dnmt1* knockout (KO) mice die at embryonic day 8.5, with the intracisternal A particle (IAP), one of the most active class II ERVs in mice, demethylated and derepressed [19]. Derepression of IAP is also observed in *Dnmt1* KO mouse embryonic fibroblasts (MEFs) and neurons, but not in mouse embryonic stem cells (mESCs), indicating the presence of DNA methylation-independent mechanisms of retroelement silencing [20]. Post-translational modifications of histone proteins regulate chromatin compaction and mediate the epigenetic regulation of transcription. Methylation of histone tails is one of the fundamental events of epigenetic signaling. In organisms ranging from the fission yeast *Schizosaccharomyces pombe* to humans, repeat-rich constitutive heterochromatin is marked by H3K9me2 or H3K9me3 [21,22,23]. These modifications are catalyzed by a family of SET-domain containing lysine methyltransferases, of which there are five in mammals. SETDB1 and the related enzymes SUV39H1 and SUV39H2 contribute to the formation of both H3K9me2 and H3K9me3 [21,24], while GLP and G9a (also called EHMT1 and EHMT2, respectively) mainly regulate H3K9me1 and H3K9me2 formation [25,26,27]. In mESCs, H3K9me3 and the presence of SETDB1 are significantly enriched in class I/II ERVs, and all retroelement types except SINEs are derepressed in *Setdb1* KO mESCs [10,28]. Moreover, 69 ERV subfamilies are derepressed in *Setdb1* KO mESCs, whereas only 5 are derepressed in *Dnmt1a/3a/3b* triple knockout mESCs, 4 of which are derepressed to a greater extent in *Setdb1* KO mESCs [28]. Thus, H3K9me3 methyltransferase SETDB1 has more central roles in ERV silencing in mESCs than DNA methylation. Although G9a/GLP and SUV39H1/2 are not involved in most of retroelement silencing, some retroelements are regulated by them. Silencing of the class III ERV MERVL depends on the catalytic activity of G9a/GLP [29], and SUV39H1/2 is responsible for H3K9me3 formation and repression of the intact L1 in mESCs [30]. Thus, the H3K9 methylation pathway is a major mechanism for retroelement silencing in mESCs, and among the various H3K9 methyltransferases, SETDB1 is the most critical retroelement silencing factor.

## 3. Characterization of SETDB1

SETDB1 was first identified as a histone methyltransferase in a screening for interaction partners for the ERG transcription factor in mice [31]. SETDB1 homologs have been found in *Drosophila* and *C. elegans* [32,33]. The human SETDB1 protein (1291 amino acids, 143 kDa) contains triple Tudor domains (TTDs) and a methyl-CpG binding domain (MBD) in its *N*-terminal part, followed by a split SET domain (pre-SET, SET, and post-SET) (Figure 1). Tudor domains are known to recognize methylated lysine and arginine [34,35]. SETDB1 TTDs bind to histone H3 tails that have been modified by K14 acetylation combined with K9 methylation, and SETDB1 is localized at H3K9me3 or H3K9me3/K14ac enriched genomic regions [36,37]. MBD is thought to mediate methyl-CpG binding and protein-protein interactions [24]. However, recombinant SETDB1 is not able to bind methylated DNA [38]. The C-terminal region containing the pre-SET, SET, and post-SET domains, which are found in the most SET family members, is responsible for H3K9 specific lysine methylation [21,24]. The SET domain of SETDB1 has a long insertion, which is monoubiquitinated by UBE2E enzymes independent of E3 ligase (on K867 in human or K885 in mouse), and this monoubiquitination highly enhances the methyltransferase activity of SETDB1 [39,40]. At the *N*-terminal region of SETDB1, there is a functional SUMO interacting motif (SIM) [41] in close proximity to a nuclear export signal (NES) and a nuclear localization signal (NLS) [42] (Figure 1). The *Drosophila* SETDB1 homolog, known as Egg or Eggless, also contains the bifurcated SET domain, TTDs, and MBD [33].

SETDB1 is expressed in various tissues (liver, brain, thymus, heart, lung, spleen, testis, ovary, and kidney) in mice [43]. During early embryogenesis, SETDB1 is also expressed in mature oocytes and primordial germ cells (PGCs). SETDB1 is required for early embryogenesis and germline development [44,45]. During early embryogenesis and germ cell development, the DNA methylation profile is dynamically changed, and DNA is hypomethylated in the blastocyst stage and PGCs [46]. Thus, SETDB1 may function as an alternative mechanism for retroelement silencing when DNA methylation is diminished. Consistent with this, SETDB1 and certain factors involved in SETDB1-mediated retroelement silencing are dramatically upregulated in mESCs cultured with 2i inhibitors (GSK3 and ERK/MEK inhibitors), which induce global DNA hypomethylation in mESCs [47].

## 4. SETDB1 Regulates Retroelements Also in Somatic Cells

It was suggested that the mechanisms of retroelement repression in differentiated adult tissues are distinct from those in ESCs or early lineage progenitors, because DNA methylation appears to be particularly important for ERV suppression, whereas histone methyltransferases responsible for H3K9me3 formation are largely dispensable [10,30]. Indeed, retroelements repressed by SETDB1 in mESCs, such as IAPs and ETns, are not activated in differentiated cells lacking SETDB1 [48]. However, recent studies of adult *Setdb1* KO mice and differentiated cells showed that SETDB1 also represses retroelements in somatic cells: B lymphocytes [49], T lymphocytes [50], brain [51], and iMEFs [52]. Consistent with this, KRAB-ZNF/KAP1, which is a pivotal factor of SETDB1 recruitment to its target regions, also functions in retroelement silencing in adult tissues [53,54,55]. Thus, SETDB1 plays a more general role in suppressing retroelements than previously thought. Although SETDB1 represses retroelements in somatic cells and differentiated cells, different retroelement types are activated by SETDB1 depletion in different cell types: VL30-class ERVs are activated in *Setdb1* KO MEFs but not in mESCs [52], and loss of SETDB1 induces robust expression of the MLV, MMTV, and VL30 ERV families in pro-B cells, whereas in *Setdb1* KO mESCs and PGCs, MMTV and VL30 remain silent and MLV are expressed at very low levels [49]. On the other hand, the IAPEz, GLn, and ETn/MusD families of ERVs are activated in ESCs and PGCs [28]. This tissue-specific derepression of retroelements by *Setdb1* KO can be explained by at least two mechanisms: regulation of tissue-specific transcription factors (TF) and tissue-specific SETDB1 targeting. In pro-B cells, only a subset of MLVs, which harbor a binding motif for PAX5, an essential B cell-lineage transcription factor, are fully de-repressed on *Setdb1* deletion, and such MLV copies are not activated by *Setdb1* KO when PAX5 is depleted [49]. Similarly, in *Setdb1* KO MEFs, derepressed copies of VL30 harbor binding motifs for TFs of the Elk family and Ets1, which are mutated in the silent or low derepressed copies. Elk-1 and Ets1 are direct targets of activated MAP kinase (MAPK), and treatment of *Setdb1* KO MEFs with MAPK inhibitors causes a strong inhibition of VL30 derepression [52]. However, frequency of K9me3 mark on retroelements that are not derepressed by depletion of these cell type specific master regulators also decreases in *Setdb1* KO in MEF and pro-B cells [49,52]. Thus, tissue-specific derepression of ERV transcription in *Setdb1* KO cells is likely caused by tissue-specific regulation of transcriptional factors, rather than tissue-specific SETDB1 targeting.

## 5. In Vivo Function of SETDB1

Zygotic SETDB1 expression begins during the blastocyst stage and is ubiquitous during post-implantation mouse development, whereas maternal *Setdb1* transcripts are present in oocytes and persist throughout pre-implantation development [44]. Homozygous mutation of *Setdb1* resulted in peri-implantation lethality between 3.5 and 5.5 dpc. Thus, *Setdb1* plays an essential role in early development [44]. SETDB1 is also essential for germ cell development [45,56,57], neurogenesis [51], osteoblast differentiation [58], chondrocyte differentiation [59,60], B-cell development [49,61], maintenance of hematopoietic stem and progenitor cells [62], T cell development [50], and restricting the differentiation potential of preadipocytes [38]. Additionally, SETDB1 regulates the expression of tissue-specific genes [38,50,51,62]. Therefore, SETDB1 plays important roles in development, cell differentiation, and other cellular functions by regulating the transcription of both genes and retroelements. Dysregulation of SETDB1 expression is found in some diseases. SETDB1 is overexpressed in many cancers. SETDB1 loss in acute myeloid leukemia (AML) cell leads to retroelement reactivation, which leads to apoptosis through a cytosolic dsRNA-sensing pathway [63]. Recently, it was reported that the decreased expression of SETDB1 in the gut leads to derepression of retroelements and ZBP1-dependent necroptosis, promoting bowel inflammation [64]. Therefore, SETDB1 is a potential therapeutic target for some diseases, and studies about SETDB1 are important.

## 6. Regulation of Subcellular Localization of SETDB1

Although SETDB1 functions in the nucleus, in various cell lines it is mainly localized in the cytoplasm [65]. Consistent with such cytoplasmic localization, SETDB1 harbors NES sequences in its *N*-terminal region [42]. Treatment with the nuclear export inhibitor leptomycin B (LMB), a CRM1-dependent nuclear export inhibitor, and the proteasome inhibitor MG132 led to the accumulation of SETDB1 in the nucleus [65], suggesting that nuclear SETDB1 might undergo degradation by the proteasome and be exported back to the cytosol. These two mechanisms might control the nuclear SETDB1 protein levels. Recently, ATF7IP (also known as mAM/MCAF1), which is a component of the SETDB1 complex and is involved in SETDB1-dependent heterochromatin formation [66,67,68], was shown to protect SETDB1 from the proteasome-mediated degradation pathway in HeLa cells [69], and regulate the nuclear accumulation of SETDB1 in mESCs and HEK293T cells [70] to maintain the nuclear SETDB1 pool. Furthermore, ATF7IP-mediated SETDB1 nuclear accumulation enhances its ubiquitinated, enzymatically more active form [70] (Figure 2). Although the detailed mechanisms underlying the function of ATF7IP in these processes remain unknown, the fact that ATF7IP interacts with the *N*-terminal region of SETDB1 (residues 1-109), which contains the two NES motifs, suggests that it might interfere with the binding of nuclear export proteins, such as CRM1, thus, increasing SETDB1 nuclear localization [70]. As expected from the marked reduction of nuclear SETDB1 protein levels in *Atf7ip* KO cells, depletion of ATF7IP mirrors the effect of SETDB1 depletion on the transcriptome [69,71]. Thus, regulation of cellular localization of SETDB1 is important for retroelement silencing. The regulation of nuclear localization of SETDB1 by ATF7IP is also conserved in *Drosophila* [72] and *Caenorhabditis elegans* [73,74]. Furthermore, the nuclear localization of Eggless/SETDB1 also increases its ubiquitination levels in ovarian somatic cells in *Drosophila* [75].

## 7. SETDB1-Mediated Retroelement Silencing: Recruitment of SETDB1 to Retroelements

It is supposed that heterochromatin formation needs the following steps: (1) de novo recruitment of a writer protein, (2) spreading of heterochromatin, and (3) maintenance of heterochromatin during cell division. Recruitment of SETDB1 to retroelements has been studied extensively. KAP1 (also called TRIM28), which forms a complex with SETDB1 and KRAB zinc finger proteins (KRAB-zfps) [24], is essential for retroelement silencing in mESCs by recruiting SETDB1 to the retroelements [10,48] (Figure 2). KAP1 harbors a RING-B box-coiled coil (RBCC) domain in its *N*-terminus and an HP1 binding domain and a tandem PHD-bromodomain in its C-terminus. The RBCC domain is responsible for Krüppel-associated box (KRAB) domain binding [78,79], while the C-terminal domains are required for gene silencing [80]. The bromodomain of KAP1 provides the essential interface for the interaction with SETDB1, which depends on the SUMOylation of several lysines of KAP1 (K554, K575, K676, K750, K779, and K804) [41,81,82]. The PHD-bromodomain module of KAP1 is necessary for the recruitment of SETDB1 to the promoter regions of genes modulated by KRAB-zfps [24]. The PHD of KAP1 functions as an intramolecular E3 ligase that SUMOylates the adjacent KAP1 bromodomain [41]. KAP1 depletion in mESCs results in decreased SETDB1 enrichment in class I/II ERVs [10]. Furthermore, the enrichment of SETDB1 in ERVs is highly correlated with that of KAP1 [29]. KAP1 and its repressive complex can be recruited to target sites by KRAB-zfps. Nearly one-third of mammalian zfps possess the highly conserved KRAB motif. Human genome sequence analysis identified 423 independent KRAB-zfp-coding loci, yielding alternative transcripts that altogether, predict at least 742 structurally distinct KRAB-zfps [83]. KRAB-zinc finger genes are one of the fastest-growing gene families and this expansion is hypothesized to enable host species to respond to newly emerged retroelements [84,85]. The coevolution between retroelements and KRAB-zfps has been reported in the case between ZNF91 and the SVA retroelement, and between ZNF93 and the L1 retroelement [86]. In addition, ChIP-exo analysis of 222 human KRAB-zfps revealed a positive correlation between the estimated evolutionary ages of KRAB-zfps and that of their transposable element targets [87]. The targeting of specific retroelements by KRAB-zfps has also been reported in mice: ZFP809 for murine leukemia virus (MLV) and its endogenous homologs [55,88], Gm6871 for LINE [89], and ZFP932 and Gm15446 for ERVK [54]. Thus, the KAP1/KRAB-zfps complex is thought as a pivotal factor for SETDB1 recruitment to retroelements (Figure 2).

## 8. SETDB1-Mediated Retroelement Silencing: Spreading of H3K9me3 from KAP1 Binding Sites

H3K9me3 often spreads from sites of KRAB/KAP1 recruitment, in some cases up to tens of Kb and represses transcription of genes located away from KAP1 binding sites [76,90,91,92]. The presence of KAP1 binding sites is restricted only to one region of the retroelement, whereas H3K9me3 enrichment occurs throughout the entire retroelement region [54,55]. Since the KRAB-zfp binding sites in retroelements do not always coincide with the binding sites of transcription factors inducing retroelement expression, H3K9me3 spreading might be important for retroelement silencing. The recruitment of SETDB1 to H3K9me3-modified genomic regions by H3K9me3-binding proteins in order to spread H3K9me3 is a simple model explaining H3K9me3 spreading. Heterochromatin protein 1 (HP1) is a representative H3K9me reader protein. There are three isoforms of HP1 in the mouse genome: HP1α (encoded by *Cbx5*), HP1β (encoded by *Cbx1*), and HP1γ (encoded by *Cbx3*) [93]. The chromodomain of HP1 is responsible for H3K9me2/3 binding [94,95]. Targeting of HP1α, HP1β and HP1γ to heterologous loci is sufficient to induce recruitment of SETDB1 and deposition of H3K9me3 [11,96]. In mESCs, ERVs are enriched in HP1α, HP1β and HP1γ and this is partially dependent on SETDB1-deposited H3K9me3 [10]. A slight derepression of ERVs and partial reduction of H3K9me3 around ERVs are observed in *HP1β* KO mESCs [11,29]. Thus, HP1 proteins are partially involved in H3K9me3 spreading. It is possible that the mild effect of HP1 depletion is due to the redundant function of HP1α, HP1β and HP1γ. Although it was reported that L1 and class II/III ERVs are derepressed in *HP1α*, *HP1β* and *HP1γ* triple KO mESCs, the resulting H3K9me3 profile was not reported [97]. Thus, the precise impact of HP1 depletion in H3K9me3 spreading remains undetermined. In addition to HP1 proteins, other chromodomain proteins, including CDYL1, CDYL2, CBX2, CBX4, CBX7, and MPP8, as well as the Tudor domain-containing protein TDRD7, were shown to bind H3K9me3 in vitro [11,98,99,100,101,102,103]. *Mpp8 (Mphospho8)* was identified by genetic screening of retroelement silencing factors in mouse and human cells [71,76,104]. MPP8 is a component of the Human Silencing Hub (HUSH) complex, which also contains FAM208A (also known as TASOR) and PPHLN1 (Periphilin1) [76]. The HUSH complex represses various viral and non-viral transgenes, driven by many different promoters, when they are integrated in H3K9me3-rich genomic regions, suggesting that the HUSH complex mediates heterochromatin spreading through H3K9me3 [76,105] (Figure 2). The HUSH complex is recruited to H3K9me3-rich genomic regions and maintains H3K9me3 by the targeted recruitment of SETDB1 [76]. The residue in the chromodomain of MPP8 that is essential for its binding to H3K9me3 is required for the efficient silencing of a reporter gene integrated in a H3K9me3-rich region [76]. Additionally, the HUSH complex is required for the enrichment of SETDB1 in HUSH targets [76]. Thus, the HUSH complex interacts with H3K9me3 via the MPP8 chromodomain and recruits SETDB1 to H3K9me3-rich regions (Figure 2). Although the HUSH complex is a candidate factor for H3K9me3 spreading, only the SETDB1-mediated H3K9me3 formation is affected by the depletion of HUSH components, and there is no direct evidence of HUSH-mediated H3K9me3 spreading. Therefore, further studies are required for unraveling the mechanism of H3K9me3 spreading within and around retroelements.

## 9. SETDB1-Mediated Retroelement Silencing: Maintenance of Heterochromatin during Cell Division

To silence retroelements stably, their association with heterochromatin should be maintained during cell division. However, it remains unknown whether the heterochromatin status of retroelements is maintained during replication or is reinstated after replication. It was reported that the replication-dependent histone assembly pathway is required for heterochromatin maintenance during reprogramming, implicating the presence of maintenance mechanisms of retroelement silencing during replication. In differentiated cells, some genomic regions with H3K9me3 form reprogramming resistant regions (RRRs) during iPS reprogramming. SETDB1 and CHAF1 are required for H3K9me3 maintenance in RRRs, and their depletion enhances reprogramming efficiency [106,107]. CHAF1 performs the first step during the chromatin-assembly process, bringing H3 and H4 to the daughter DNA strands [108]. CHAF1 interacts with HP1 and SETDB1, and is required to ensure the formation of a replication-specific pool of HP1 molecules at the replication sites of pericentric heterochromatin during the mid-late S phase [109]. Moreover, CHAF1 is essential for heterochromatin organization in pluripotent embryonic cells [110], and both CHAF1 and its upstream histone chaperones ASF1A/B are required for efficient provirus silencing in mESCs [111]. Thus, it is possible that CHAF1-mediated histone assembly has an important role in the maintenance of the heterochromatin status of retroelements during replication.

## 10. Retroelement-Type Specific Regulation

In mESCs, SETDB1 regulates the expression of class I/II/III ERVs and L1 retroelements. Recent gene knockdown or knockout screens followed by transcriptome analysis revealed that SETDB1 cooperates with different protein complexes to repress distinct retroelement types [71,111]. Yang et al. performed an siRNA knockdown screen in F9 embryonic carcinoma cells using a provirus *MMLV-Gfp* reporter gene and identified 303 candidate genes as provirus silencing factors. Among these candidates, they focused on the SETDB1 complex (*Setdb1, Kap1,* and *Atf7ip*), the CHAF1 complex (*Chaf1a* and *Chaf1b*), and the SUMOylation modification complex (*Sumo2, Sae1, Ube2i, Ube2,* and *Senp6*). Fukuda et al. performed a CRISPR-based gene knockout screen in mESCs and identified more than 100 genes as MSCV provirus silencing factors; they focused on the SETDB1/ATF7IP complex, the HUSH complex, the ATRX/DAXX H3.3 histone chaperone complex, and the newly identified retroelement silencing factors MORC2A and RESF1. Transcriptome analysis in cells depleted for each of these factors revealed which retroelement type was regulated by each protein complex.

SETDB1 complex: The pattern and degree of retroelement derepression upon SETDB1 or ATF7IP depletion were highly correlated, suggesting a central role of ATF7IP in SETDB1-mediated retroelement silencing. The detailed function of ATF7IP has already been discussed in a previous section (Figure 2).

SUMOylation modification pathway complex: Among the components of this complex, SUMO2 depletion led to the most significant induction of retroelement activation, suggesting a central role for SUMO2 in this SUMOylation process. SUMO2 represses class I/II ERVs targeted by SETDB1, such as VL30 (class I) and MMERVK10C (class II). As mentioned above, SUMOylation of KAP1 is required for the KAP1-SETDB1 interaction. Consistent with this, the genomic binding pattern of SUMO2 highly overlaps with that of the SETDB1 complex. Enrichment of SUMO2 in the KAP1 target genomic regions depends on KAP1, and SUMO2 is required for the efficient enrichment of H3K9me3 in those regions. In addition, inhibition of the SUMOylation pathway or depletion of hnRNPK, which interacts with the SETDB1/KAP1 complex and regulates the SUMOylation levels of ERV loci by an unknown mechanism, results in a decrease in SETDB1 enrichment in ERVs [91]. Thus, it is hypothesized that the SUMOylation modification pathway plays critical roles in ERV silencing via the recruitment or retention of SETDB1 through SUMOylation of proteins such as KAP1 on ERV chromatin. It has been shown that some class III ERVs, such as MERVL, are repressed by KAP1 in an SETDB1-independent manner [29]. SUMO2 also represses KAP1-dependent and SETDB1-independent class III ERVs, suggesting the presence of KAP1 SUMOylation functions other than the interaction with SETDB1. Since KAP1 SUMOylation is required for its interaction with CHD3, which is a component of the NuRD complex [41], and HDAC, which is involved in MERVL repression [112], KAP1 SUMOylation might function in HDAC recruitment to class III ERVs (Figure 3A).

CHAF1 complex: In mESCs, CHAF1A/B represses a larger number of class III than class I/II ERVs. CHAF1A-targeted class III ERVs are not marked with H3K9me3, but instead are enriched in H3K4me2 and H3K27Ac, which are targets of histone demethylase KDM1A and histone deacetylase HDAC2. CHAF1A-targeted class III ERVs are also enriched in both these enzymes, which interact with CHAF1A. On the other hand, CHAF1A targeted class I/II ERVs with high H3K9me3 levels are not enriched in KDM1A/LSD1 and HDAC2. CHAF1A depletion results in increased H3K4me3 and H3 acetylation of CHAF1A-targeted class III ERVs. CHAF1A represses ERVs also by cooperating with KAP1 to increase the levels of H3K9me3 in class I and II ERVs. In addition, CHAF1 depletion results in a significant increase of active chromatin modification in proviruses, whereas H3K9me3 is only slightly decreased. Thus, CHAF1A regulates class I/II and III ERVs by different mechanisms through interactions with distinct co-factors [111] (Figure 3A,B).

The ATRX/DAXX complex: This complex incorporates the histone variant H3.3 at heterochromatin regions in a replication-independent manner. ATRX contains an ATPase/helicase domain and belongs to the SWI/SNF family of chromatin remodeling proteins. In mESCs, class I/II ERVs become enriched in histone H3.3 in an ATRX/DAXX-dependent manner, through the act of DAXX acts as histone chaperone that facilitates the deposition of H3.3 [114,119]. Although the role of H3.3 in ERV silencing seems to be minimal [114,119,120], ATRX and DAXX depletion leads to the derepression of class I/II ERVs [71,114,121]. Recruitment of DAXX, H3.3, and KAP1 to ERVs is co-dependent and occurs upstream of SETDB1, linking H3.3 to ERV-associated H3K9me3. However, *Daxx* KO in mESCs has a higher impact on ERV depression than *H3.3* KO and according to H3.3 ChIP-seq experiments in wildtype mESCs, H3.3 is not enriched at some ERVs that become derepressed in H3.3 KO mESC [119], implicating functions of H3.3 other than incorporation into nucleosomes. Interestingly, the DAXX protein levels are decreased in *H3.3* KO mESCs, suggesting a role of H3.3 for DAXX stabilization [115]. In addition, ERV suppression in *H3.3* KO mESCs can be rescued by an H3.3 mutant that is not incorporated in nucleosomes but still interact with DAXX, but not by an H3.3 mutant that does not interact with DAXX [115]. DAXX also forms a complex with HDAC1, SETDB1, and KAP1, and the interaction between HDAC1 and KAP1 is dependent on DAXX. In addition, ERV derepression caused by treatment with the HDAC inhibitor TSA is diminished in *Daxx* KO mESCs [115]. Therefore, the role of H3.3 in retroelement silencing is likely to stabilize DAXX in order to maintain the interaction between KAP1 and HDAC1, rather than involving its incorporation into nucleosomes. Because ATRX is not included in the DAXX-SETDB1-KAP1-HDAC1 complex and some ERVs, such as IAP and MusD, are derepressed in *Daxx* KO mESCs but not in *Atrx* KO mESCs, the function of DAXX mentioned above is ATRX-independent [115] (Figure 3B).

Although IAP is not derepressed in *Atrx* KO mESCs, IAP regions become more accessible. Similarly to H3K9me3 and H3.3, ATRX is enriched in the entire region of IAP, and this enrichment might be mediated by the H3K9me3 binding ability of the ADD domain of ATRX [121,122]. Thus, ATRX/DAXX is required for efficient heterochromatin compaction in ERVs (Figure 3). In contrast with *Atrx* KO in mESCs, temporal depletion of ATRX leads to IAP derepression [47,71], implicating the presence of a compensation mechanism against long-term ATRX loss in mESCs. Since IAP derepression by KAP1 knockdown is more significant in *Atrx* KO mESCs than in WT mESCs [121], KAP1 might be involved in this compensation mechanism. We found a high degree of overlapping between the targets of the newly identified retroelement silencing factor RESF1 and those of ATRX/DAXX. Although RESF1 interacts with SETDB1 and stabilizes SETDB1 enrichment in proviruses, it remains unknown if RESF1 associates with the ATRX/DAXX complex.

HUSH complex: In contrast to the SETDB1/ATF7IP complex, the HUSH complex strongly represses L1, especially young and full-length L1, rather than ERVs in mESCs. Regulation of L1 by the HUSH complex was also reported in humans [104]. The HUSH complex mainly represses L1, and full-length and young L1 is enriched in MPP8 in mESCs and K562 cells [104]. Similar to the HUSH complex, both human MORC2 and its mouse homolog, MORC2A, also repress young and full-length L1 in K562 cells and mESCs [71,104]. In humans, MORC2 interacts with the HUSH complex and is involved in HUSH complex-mediated transcriptional silencing [104]. A certain *MORC2* mutation that hyperactivates the HUSH complex causes axonal Charcot–Marie–Tooth disease [117]. These results suggest that MORC2A modulates HUSH complex functions. Interestingly, in K562 cells and hESCs, MORC2 and the HUSH complex target young, full-length L1s, especially when they are located within introns of actively transcribed genes, and L1 transcription increases MORC2 and MPP8 occupancy at the L1 transgene [104]. Thus, the HUSH complex and MORC2 preferably target L1s presenting the highest potential threat to genome integrity. How does the HUSH complex target transcribed L1s? Recently, it was reported that the disordered *N*-terminal domain of PPHLN binds RNA non-specifically and forms insoluble aggregates, whereas its structured C-terminal domain binds TASOR. Moreover, residues required for both RNA aggregation and TASOR binding are also required for HUSH-mediated silencing [116]. Although it remains unknown whether PPHLN binds to nascent L1 transcripts, it is possible that PPHLN stabilizes the accumulation of the HUSH complex on transcribed L1 by its RNA binding ability (Figure 3C). The next question is how MORC2 regulates HUSH targets. MORC2 harbors GHKL-type ATPase, S5, CW and chromo-like domains, and multiple coiled-coil domains (CC1, CC2, and CC3). All domains except for the chromo-like domain are required for HUSH-mediated silencing in HeLa cells [117]. Its ATPase and CW domains are also required for provirus silencing in mESCs [71]. The MORC2 ATPase module dimerizes upon ATP binding, which is required for silencing. MORC2 binds both free dsDNA and nucleosomal DNA, and CC1 contributes to the MORC2 DNA-binding activity [118]. As MORC2 enrichment in heterochromatin regions depends on the HUSH complex, and loss of MORC2 leads to chromatin decompaction in HUSH-target sites in HeLa cells [117], MORC2 might be recruited by the HUSH complex to chromatin, where dimerization of two DNA-bound MORC2 can promote DNA loop formation and chromatin compaction [118] (Figure 3C). In addition to chromatin compaction, MORC2 might induce histone deacetylation of L1s by recruiting HDAC, since MORC2 association with HDAC1/4 is required for gene silencing [123,124].

## 11. Concluding Remarks

To understand heterochromatin formation on retroelements, it is necessary to unravel the following mechanisms: (1) de novo recruitment of a writer protein, (2) heterochromatin spreading, and (3) maintenance of heterochromatin during cell division. It has been revealed that the KRAB-zfps/KAP1 pathway is central for the de novo recruitment of SETDB1 to retroelements. In mESCs, conditional *Kap1* KO caused less derepression of certain ERVs than *Setdb1* KO. This difference might be due to different mechanisms for the maintenance of H3K9me3. However, in contrast to the mechanism of de novo recruitment of SETDB1, spreading mechanisms of SETDB1-mediated H3K9me3 are poorly understood. Although the HUSH complex is a candidate that may regulate the spreading, SETDB1-mediated H3K9me3 is only partially regulated by the HUSH complex [76]. This can be also explained by the presence of maintenance mechanisms of SETDB1-mediated H3K9me3. Although it remains unknown whether SETDB1-mediated heterochromatin is maintained during replication or is reinstated after cell division, the involvement of CHAF1 in SETDB1-mediated heterochromatin formation implicates the presence of maintenance mechanisms during replication. Further studies are required to reveal how CHAF-1-mediated histone depletion regulates heterochromatin maintenance.

Recently, many genes have been identified as retroelement silencing factors. However, it remains poorly understood which step of heterochromatin formation is regulated by which factor. Thus, it is important to gain such knowledge in order to understand retroelement silencing and heterochromatin formation.

## Figures and Tables

**Figure 1 viruses-12-00596-f001:**
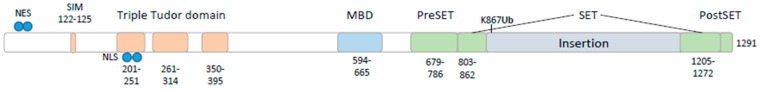
Protein domain structure of SETDB1. The human SETDB1 protein (1291 amino acids, 143 kDa) contains triple Tudor domains (TTDs) and a methyl-CpG binding domain (MBD) in its *N*-terminal part, followed by a split SET domain (pre-SET, SET, and post-SET). The TTDs bind to histone H3 tails that have been modified by K14 acetylation combined with K9 methylation. The C-terminal region containing the pre-SET, SET, and post-SET domains is responsible for H3K9 specific lysine methylation [21,24]. The SET domain of SETDB1 has a long insertion, which is monoubiquitinated by UBE2E enzymes independent of E3 ligase (on K867 in human or K885 in mouse), and this monoubiquitination highly enhances the methyltransferase activity of SETDB1 [39,40]. At the *N*-terminal region of SETDB1, there is a functional SUMO interacting motif (SIM) [41] in close proximity to a nuclear export signal (NES) and a nuclear localization signal (NLS) [42].

**Figure 2 viruses-12-00596-f002:**
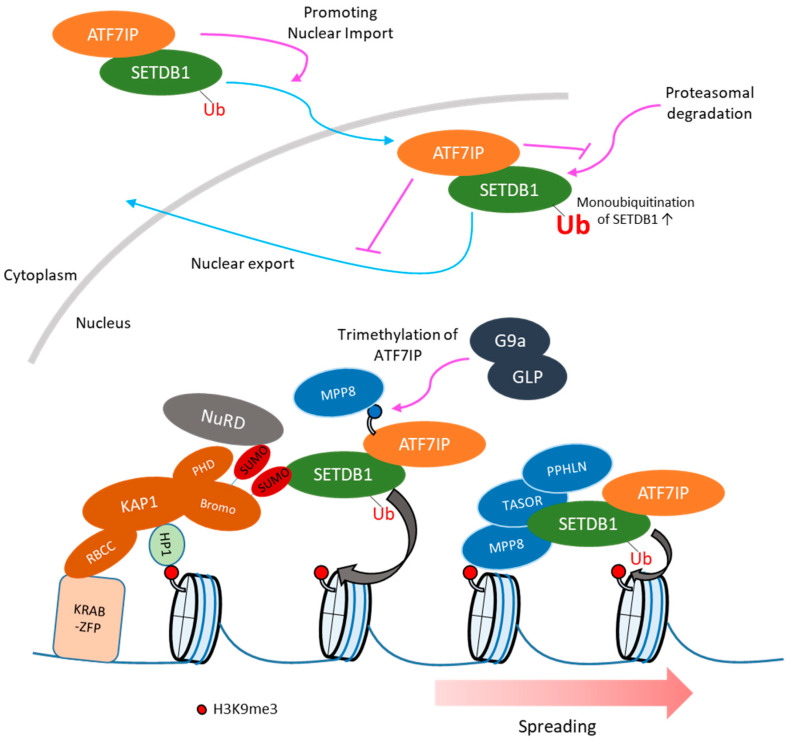
SETDB1 recruitment and H3K9me3 spreading. SETDB1 is mainly localized in cytoplasm and nuclear SETDB1 is degraded by proteasomal pathway [65]. ATF7IP promotes nuclear import of SETDB1 and inhibits nuclear export and proteasomal degradation of SETDB1 [70]. In addition, ATF7IP-mediated SETDB1 nuclear accumulation enhances its ubiquitinated, enzymatically more active form [70]. SETDB1 interacts with SUMOylated KAP1 and is recruited to retroelement by KRAB-ZFP/KAP1 pathway, then SETDB1 mediates H3K9me3 in retroelement [41]. The Human Silencing Hub (HUSH) component bind H3K9me3 via chromodomain of MPP8, which might require for H3K9me3 spreading [76]. ATF7IP harbors H3K9-like sequence, which is methylated by G9a/GLP. The methylation of ATF7IP is recognized by MPP8 and the residue methylated by G9a/GLP is required for efficient provirus silencing [77]. The HUSH complex might be recruited to SETDB1 target via interaction between methylated ATF7IP and MPP8.

**Figure 3 viruses-12-00596-f003:**
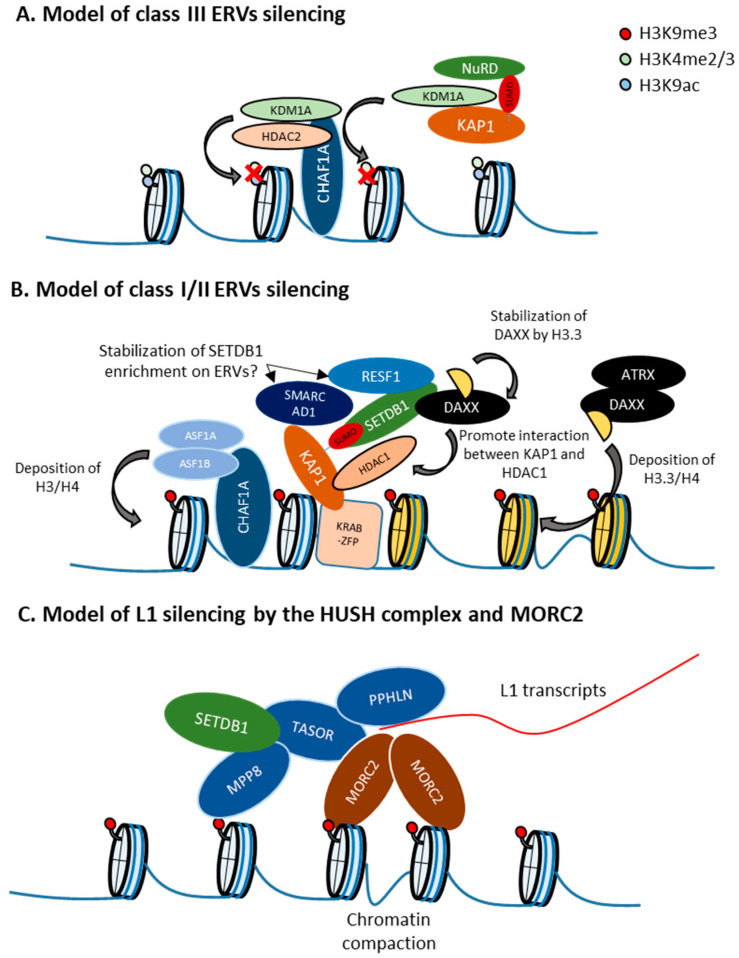
Retroelement-type specific regulation. (**A**) Model of class III endogenous retroviruses (ERVs) silencing. CHAF1A recruits KDM1A and HDAC2 to class III ERVs to remove active chromatin marks such as H3K4me2/3 and H3K9ac [111]. KAP1 and SUMO2 also represses SETDB1-independent class III ERVs. Since KAP1 SUMOylation is required for its interaction with CHD3, which is a component of the NuRD complex [41], and HDAC, which is involved in MERVL repression [112], KAP1 SUMOylation might function in HDAC recruitment to class III ERV. (**B**) Model of class I/II ERVs silencing. Recruitment of SETDB1 by interaction with SUMOylated KAP1. SMARCAD1 and RESF1 might enhance SETDB1 enrichment in class I/II ERVs [71,113]. ATRX/DAXX deposits H3.3 to class I/II ERVs and play a role for silencing of such ERVs [114]. H3.3 stabilizes the DAXX protein to promote interaction between KAP1 and HDAC1 [115]. Replication-dependent H3/H4 deposition by CHAF1 is also involved in provirus and class I/II ERVs silencing [111]. (**C**) Model of L1 silencing by the HUSH complex and MORC2. MORC2 and the HUSH complex target young, full-length L1s, especially when they are located within introns of actively transcribed genes, and L1 transcription increases MORC2 and MPP8 occupancy at the L1 transgene [104]. The disordered *N*-terminal domain of Periphilin1 (PPHLN) might bind RNA non-specifically and forms insoluble aggregates [116], which might enhance the HUSH complex targeting to transcriptionally active L1. MORC2 interacts with the HUSH complex and MORC2 enrichment at HUSH targets is dependent on the HUSH complex [117]. The MORC2 ATPase module dimerizes upon ATP binding and MORC2 binds both free dsDNA and nucleosomal DNA [118]. MORC2 might be recruited by the HUSH complex to chromatin, where dimerization of two DNA-bound MORC2 can promote DNA loop formation and chromatin compaction [118].

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
