# Peer review of "SETDB1-Mediated Silencing of Retroelements"

_viruses, 2020, doi:10.3390/v12060596_

Round 1
Reviewer 1 Report
Comments
Overall this is a nice review about the role of SETDB1 in repressing retroelements, that takes into consideration the most up to date research on this topic. It is scholarly, authoritative, and generally well written. I have a number of minor suggestions.
- First sentence of abstract in misleading. What is “the mammalian genome”? Perhaps it’s more accurate to say that they comprise greater than 40% of most extant mammalian genomes.
- The introduction paragraph is identical to the abstract.
- The authors state that “ERVs are abundant in mammals, comprising approximately 8% of the mouse genome and 10% of the human genome”. The data is 10% of mouse and 8% of humans.
- The authors state that “Thus, tissue-specific derepression of ERV transcription in Setdb1 KO cells might be caused by tissue-specific regulation of transcriptional factors, rather than tissue-specific SETDB1 targeting.” But the data reported seems pretty convincing. I would suggest saying “is likely caused” instead of “might be caused”.
- The authors state “However, H3K9me3 of retroelement without derepression is also decreased by Setdb1 KO in MEFs and pro-B cells (48,51)”: this sentence was hard to read. Maybe “However, frequency of K9me3 mark on retroelements that are not derepressed by depletion of these cell type specific master regulators also decreases in Setdb1 KO in MEF and pro-B cells.”
- Authors state “Zygotic Setdb1 expression begins during blastocyst stage, ubiquitous during post-imp, maternal setdb1 transcripts are present in oocytes and persist through pre-implantation”: please add sources for these two statements.
- The authors state “The arms race between retroelements and KRAB-zfps has been reported in the case between ZNF91 and the SVA retroelement, and between ZNF93 and the L1 retroelement.” The authors should explain what is meant by the arms race since it will not be apparent to many readers.
- The authors state “The generation of H3K9me3 is spread in several tens of kbs away from the KAP1 binding sites and represses transcription of genes located away from KAP1 binding sites (87-90).” However this is not always the case. At many ERVs there is little spreading of H3K9me3. Thus I think the authors should provide a more nuanced view. Perhaps they can state that H3K9me3 often spreads from sites of KRAB/KAP1 recruitment, in some cases up to tens of Kb.
- Citations to the Elsasser papers (112 and 113) should include citation of the Nature Brief Communication article (Wolf, G et al and Faulkner, Mager, Lorincz, Macfarlan) that challenges several of the key findings and more thoroughly describes the changes upon conditional Daxx KO (as opposed to the constitutive KO used by Elsasser.). This should be discussed.
Reviewer 2 Report
Comments for Authors:
This manuscript, submitted by Fukuda and Shinkai, describes recent progress of research on the roles of SETDB1 and related proteins in silencing mammalian retroelements such as ERVs and L1. The manuscript is well written and will help readers to understand the recent advancements in this field. I consider this paper is suitable for publication in Viruses. However, I found a few points which the authors need to address.
1) The Introduction paragraph is a repetition of the Abstract. In general the contents of Abstract and Introduction may be partly overlapped but should be different even in such a review article. This manuscript describes the silencing models for L1 and different classes of ERVs that involve many proteins such as KAP1 and HUSH complex, and it would be better to add this point in the Abstract.
2) Page 2, lines 2-3; "The former are also referred to as ERVs, ..."
In general, ERVs are not equal to LTR retrotransposons but just a group among them. For example, Ty1/Copia is a kind of LTR retrotransposons but not an ERV. In the case of mammals, however, ERVs occupy an almost entire fraction of LTR retrotransposons although a small proportion of LTR retrotransposons consists of Gypsy-type elements.
This sentence should be changed to something like "The former includes ERVs, ..." if the sentence means the general classification of LTR retrotransposons.
3) Page 3, line 17;
"and germline development (43) (44)" --> "and germline development (43,44)"
4) Page 7, the first sentence of 10. Retroelement-type specific regulation.
"the expression of class I/II/III" --> "the expression of class I/II/III ERVs"
